# Diverse Physiological Functions of Cation Proton Antiporters across Bacteria and Plant Cells

**DOI:** 10.3390/ijms21124566

**Published:** 2020-06-26

**Authors:** Masaru Tsujii, Ellen Tanudjaja, Nobuyuki Uozumi

**Affiliations:** Department of Biomolecular Engineering, Graduate School of Engineering, Tohoku University, Aobayama 6-6-07, Sendai 980-8579, Japan; masaru.tsujii.c4@tohoku.ac.jp (M.T.); ellen.c7@tohoku.ac.jp (E.T.)

**Keywords:** ion transporter, cation proton antiporter, bacteria, cyanobacteria, plant

## Abstract

Membrane intrinsic transport systems play an important role in maintaining ion and pH homeostasis and forming the proton motive force in the cytoplasm and cell organelles. In most organisms, cation/proton antiporters (CPAs) mediate the exchange of K^+^, Na^+^ and Ca^2+^ for H^+^ across the membrane in response to a variety of environmental stimuli. The tertiary structure of the ion selective filter and the regulatory domains of *Escherichia coli* CPAs have been determined and a molecular mechanism of cation exchange has been proposed. Due to symbiogenesis, CPAs localized in mitochondria and chloroplasts of eukaryotic cells resemble prokaryotic CPAs. CPAs primarily contribute to keeping cytoplasmic Na^+^ concentrations low and controlling pH, which promotes the detoxification of electrophiles and formation of proton motive force across the membrane. CPAs in cyanobacteria and chloroplasts are regulators of photosynthesis and are essential for adaptation to high light or osmotic stress. CPAs in organellar membranes and in the plasma membrane also participate in various intracellular signal transduction pathways. This review discusses recent advances in our understanding of the role of CPAs in cyanobacteria and plant cells.

## 1. Introduction

The flux of cations and protons across membranes is a key process in maintaining intracellular ion homeostasis in various environments. A large number of ion transporters in the plasma membrane and in organellar membranes are involved in the regulation of ion translocation. They mediate the flux of potassium ions (K^+^), sodium ions (Na^+^) and protons (H^+^), common cations in cells and in the environment. The concentration of K^+^ in cells is estimated to be around 0.1-0.2 M, and it is the most prevalent cation in living cells [1,2]. K^+^ is required for many biological processes such as the formation of the membrane potential, osmotic adjustment and turgor pressure [3,4,5,6]. K^+^ is also needed for the proper activation of certain enzymes, for instance pyruvate kinase [7]. In contrast, the Na^+^ concentration in cells is usually only 5-14 mM. Excess Na^+^ often has negative effects on terrestrial plants and some bacteria [8,9,10,11] since a Na^+^/K^+^-ATPase is missing in bacteria and plants [12]. Excess Na^+^ induces dehydration and reduces the intracellular K^+^ content, resulting in fatal cell damage.

The exchange of K^+^ or Na^+^ with H^+^ across cell membranes and organelles is coordinated by a variety of ion transporters. Among them, the cation/proton antiporter (CPA) superfamily is one of the larger groups in the superfamily of ion transporters in a variety of organisms [13]. Transport direction and localization of CPAs are diverse and CPAs therefore play a range of physiological roles. For example, some CPAs respond to salt stress by exporting excess cations from the cytoplasm into the extracellular space or into organelles [14,15,16]. Some CPAs are involved in the transport of proteins into endosomal compartments [17,18].

Plants contain a large number of conserved CPAs [19], which are localized in the plasma membrane and in the membranes of organelles such as vacuoles, endosomes and chloroplasts. Because mitochondria and chloroplasts evolved from prokaryotes through symbiogenesis, understanding the physiological role of prokaryotic CPAs is important to elucidate their role in plants. In this study, we focus on the fundamental structure and physiological roles of CPAs in *Escherichia coli*, cyanobacteria and plant cells to evaluate their similarities and differences.

## 2. Structure of CPAs

While numerous studies have shown the physiological importance of cation/H^+^ antiporters, their structure and transport activity remain poorly understood. Direct measurement of cation/H^+^ exchange activity by CPAs is usually conducted using inverted membranes of *E. coli,* organelles of eukaryotic cells or liposomes [15,20,21,22,23] (Figure 1A). Fundamental understanding of CPAs in living cells comes from crystallography studies of the *E. coli* transporter NhaA (henceforth NhaA). NhaA, an electrogenic pH-dependent Na^+^(Li^+^)/H^+^ antiporter [24], is a member of the CPA superfamily, from the NHA clade of CPA2 [13]. To date, two different crystal structures of NhaA have been reported, one in monomeric [25] and one in dimeric configuration [26]. Both structures have been determined under low pH conditions (at pH 4 and 3.5), and therefore they correspond to the inactive state of NhaA [25,26]. NhaA is composed of 12 transmembrane segments (TMs) with N and C terminus located in the cytoplasm [25]. The monomeric crystal structure of NhaA reveals a unique structural conformation consisting of TM4 and TM11 [25], designated as the NhaA fold [27]. These two transmembrane regions contain the ion-binding site near their intersection [27,28]. The NhaA fold is critical for antiporter activity by creating an electrostatic balance in close proximity to the substrate binding site [25,28]. The Na^+^/Li^+^ binding site is composed of two essential aspartic acid residues (D163 and D164) in TM5, together with D133 and T132 in TM4 (Figure 1B) [25], which was confirmed by isothermal calorimeter experiments [29]. The crystal structure also reveals a putative ‘pH sensor’ region, a cluster of ionizable residues located at the opening of the cytoplasmic funnel (N-terminal region of TM9) [25]. Mutations affecting amino acid residues in the pH sensor region and in the TM8-9 loop significantly change the pH profile of the antiporter. Increasing pH induces conformational changes (resulting in movement of the TM4 and TM11 assembly) that expose the Na^+^ binding site and eventually activate NhaA [28].

The stoichiometry of NhaA is 1Na^+^/2H^+^ but the exact mechanisms of the antiport activity is still under debate, due to unavailability of an active-state structure. In particular, the identity of the second H^+^ donor remains unclear. Both D163 and D164 have been proposed as the first and second H^+^ donors in the NhaA antiporter cycle [25]. However, the second, dimeric NhaA crystal structure reassigns TM10, which changes the position of K300 to be within salt bridge distance to D163, which suggests an alternative hypothesis of K300 as the second H^+^ donor (Figure 1C) [26]. Furthermore, the constant pH molecular dynamics simulations (CpHMD) suggest that Na^+^ binding to D163 disrupts the salt bridge between D163 and K300, destabilizes the charged state of K300 and sparks the discharge of H^+^ [30]. Consequently, CpHMD analysis supports the proposed mechanism of K300 as the second H^+^ donor [30]. Interestingly, a recent report shows that replacement of both salt bridge residues (D163N/K300Q) does not abolish electrogenic transport activity despite the absence of a salt bridge near the ion binding site. The authors concluded that NhaA might have several alternative mechanisms to ensure the electrogenicity of the Na^+^/H^+^ exchange and that NhaA is able to utilize an alternative second H^+^ donor when the main second H^+^ donor is unavailable [31]. Therefore, efforts to solve the NhaA active structure are still ongoing.

Regardless of differences in amino acid sequence and length, some other members of CPA have a similar structural conformation to NhaA, such as *Thermus thermophilus* NapA (TtNapA), *Neisseria meningitidis* ABST (NmABST), *Pyrococcus abyssi* NhaP (PaNhaP), and *Methanocaldococcus jannaschii* NhaP1 (MjNhaP1) [27]. A broad phylogenic analysis of 6000 CPAs found a common CPA motif, consisting of eight non-linear amino acids that might be responsible for electrogenicity and ion selectivity of the CPA superfamily [32]. The CPA motif is located in the vicinity of the ion binding site [32]. The electrogenicity of the transport activity of an electrogenic CPA is usually characterized by triplet residues (two acidic residues corresponding to D163 and D164 at NhaA and a basic residue corresponding to K300 at NhaA) [32].

## 3. Physiological Functions of CPAs in *E. coli*

### 3.1. CPAs as Na^+^ Extrusion Systems

Excess Na^+^ has detrimental effects on *E. coli* growth. *E. coli* contains eight CPAs (Table 1 and, Figure 2A). Of those, three Na^+^ extrusion systems, NhaA, NhaB and ChaA, help to maintain a low concentration of Na^+^ in the cytoplasm. NhaB (formerly AntB) was discovered by analyzing the remaining antiporter activity in an *nhaA* deletion mutant [33]. NhaA and NhaB share relatively low homology of their amino acid sequence (22.4% identity and 35.6% similarity) [34]. NhaB has distinct characteristics from NhaA in terms of weak pH-dependency and lower Li^+^ affinity (approximately 15 times lower than NhaA) [33]. The third system, ChaA, was initially thought to be a Ca^2+^/H^+^ antiporter in *E. coli.* However, a follow-up study shows that ChaA has Na^+^ extrusion activity that is able to complement *E. coli* EP432 (*nhaA nhaB*) [35]. An *E. coli* mutant lacking all three of these Na^+^ extrusion systems lacks most Na^+^ extrusion ability, evidenced by its sensitivity to 0.1 M NaCl at pH 8.5 [36]. NhaA, NhaB and ChaA prevent Na^+^ toxicity under physiological conditions.

Why does *E. coli* have multiple Na^+^ extrusion systems (Figure 2A)? The answer probably lies in the different optimal pH conditions for each transporter. NhaA is strongly pH dependent and is only active at pH between 6.5 and 8.5, NhaB functions at pH below 8 and ChaA is active above pH 8 [36]. The mechanism leading to pH activation of NhaB and ChaA remains to be determined. Another characteristic differing between NhaA, NhaB and ChaA is their substrate specificity. NhaA and NhaB transport Li^+^ in addition to Na^+^, which promotes Li^+^ detoxification [37]. ChaA has a broader range of substrates: Na^+^, Ca^+^, and K^+^. Although ChaA has Ca^+^/H^+^ antiport capacity, ChaA does not play a role in Ca^2+^ extrusion in vivo [38], but rather is involved in protection against K^+^ salinity stress [39].

### 3.2. CPAs as pH Regulators at Alkaline pH

The extrusion of Na^+^ from cells via CPAs is accompanied by H^+^ uptake, which results in cytoplasmic acidification. *E. coli* is a neutralophilic bacterium which needs to maintain an intracellular pH of around 7.4–7.8 over a wide range of external pH from 5.0 to 9.0 [40]. At alkaline pH conditions, pH homeostasis can be achieved through the electrogenic nature of NhaA and NhaB transport. NhaA transports 2H^+^ per 1Na^+^ exported [41], whereas NhaB transports 3H^+^ per 2Na^+^ exported [42]. Therefore, H^+^ accumulation leads to an acidification of the cytoplasm at alkaline pH in the presence of Na^+^. Besides those three Na^+^/H^+^ transporters, two other transport systems also assist in maintaining pH homeostasis in *E. coli*: MdtM and MdfA. Both transporters are multidrug-resistance transporters (Mdr transporter), that mainly function as multi-drug/H^+^ antiporters but also have cation/H^+^ antiport activity. Lewison et al. [43] found that MdfA also mediates Na^+^(K^+^)/H^+^ antiport, which contributes to alkali tolerance in *E. coli* at extremely high pH (9.5–10). In addition, MdtM, an MdfA homolog, also exhibits Na^+^ (K^+^, Rb^+^, Li^+^)/H^+^ antiport capacity [44]. Both MdtM and MdfA are able to extend the pH tolerance of *E. coli* up to pH 10 in the presence of Na^+^ or K^+^. Since ChaA, MdtM and MdfA have K^+^/H^+^ antiport capacity, they are able to conduct H^+^ influx at higher alkaline pH conditions than Na^+^(Li^+^)-specific antiporters (NhaA and NhaB). Their ability to utilize K^+^ as a driving force is a critical advantage in adaptation to extreme alkaline conditions, since *E. coli* accumulates high K^+^ concentration intracellularly [45].

### 3.3. CPAs as Antimicrobial Stress Mechanisms 

Another physiological function of CPAs in *E. coli* is antimicrobial resistance. This function can be further divided into two categories: direct and indirect. CPAs, like MdtM and MdfA, directly contribute to antimicrobial resistance by actively extruding a broad range of toxic compounds such as antibiotics, mutagen and xenobiotics: chloramphenicol, ethidium bromide, various quaternary ammonium compounds and unconjugated bile salt by MtdM [46,47,48], zwitterionic lipophilic compounds, certain aminoglycosides and fluoroquinolones by MdfA [49].

*E. coli* takes up K^+^ through Kup, Kdp, TrkG and TrkH [50], but effluxes K^+^ via KefB and KefC under electrophilic stress conditions. KefB and KefC indirectly contribute to electrophilic stress tolerance as they transport H^+^ in exchange with K^+^ to acidify the cytoplasm which in turn alleviates the stress [51,52]. They are glutathione-gated K^+^ efflux systems in *E. coli*; each efflux system requires an ancillary protein: KefG (formerly YabF) for KefB and KefF (formerly YheR) for KefC, respectively, to stabilize their conformation and achieve maximum activity [53]. KefC consists of a transmembrane domain and a cytosolic KTN (K^+^ transport nucleotide binding) domain. The KTN domain is involved in dimer formation, and binding of glutathione to its cleft inhibits KefC activity [54,55]. When cells are exposed to electrophiles (e.g., methylglyoxal, *N*-ethylmaleimide and chlorodinitrobenzene), these react with glutathione to form glutathione conjugates which in turn activate KefB and KefC-mediated K^+^/H^+^ antiport transport activity [52].

YcgO, also known as CvrA, contributes to cell volume regulation under low osmolarity conditions [56]. YcgO functions as K^+^ efflux system in a mutant lacking PtsN (EIIANtr) [57]. YgcO activity is regulated by the phosphorylation state of PtsN, a member of the nitrogen phosphor transferase system (PTSNtr). It is therefore possible that certain stresses, that shift the balance toward phospho-PtsN, activate K^+^ efflux via YgcO and induce growth cessation [57].

## 4. Physiological Function of CPAs in Cyanobacteria

Cyanobacteria, the ancestors of plant chloroplasts, need to control the Na^+^ content in their cytoplasm. Salt stress decreases cell volume and stimulates the inactivation of photosystem I (PSI) and photosystem II (PSII) [58]. Na^+^/H^+^ antiporters are thought to be the conduit of Na^+^ influx and efflux in cyanobacteria. *Synechocystis* sp. PCC6803 (hereafter *Synechocystis*), the model strain of cyanobacteria, possesses six Na^+^/H^+^ antiporters (NhaS1-NhaS6) (Table 2 and Figure 2B) [16,20,59,60,61,62]. Among them, only NhaS1 and NhaS3 were shown to be functional Na^+^/H^+^ antiporters when expressed in *E. coli* [20,59], whereas the activity of other antiporters remains elusive. A triple mutant of NhaS1/4/5 and a single mutant of NhaS1/2/4/5 do not show any significant difference compared with the wild type, suggesting that these transporters are not involved in salinity stress adaptation [16,60]. On the other hand, Δ*nhaS3* has a salt-sensitive phenotype, suggesting that NhaS3 confers salt tolerance [16,20]. NhaS3 transports Na^+^ from the cytosol to the thylakoid lumen to keep concentrations of Na^+^ low in *Synechocystis* [20]. Freshwater cyanobacterium, *Synechococcus* sp. strain PCC7942 (hereafter *Synechococcus*) possesses seven Na^+^/H^+^ antiporters (Table 2). *Synechococcus* NhaS3 alleviates salinity stress in a pH-dependent manner [63], similar to the role of *Synechocystis* NhaS3. Loss of *nhaS2* (Δ*nhaS2*) reduces growth in media containing low Na^+^ concentrations [61] and deletion of *nhaS4* also results in hypersensitivity to low Na^+^ conditions [16]. While Δ*nhaS2* requires 250 mM NaCl in the medium to reach growth that is similar to the wild type, Δ*nhaS4* only requires 20 mM NaCl. NhaS2 and NhaS4 act as Na^+^ influx transporters with different affinities for Na^+^.

In eukaryotes, Ca^2+^ is an essential cation, however, the physiological role of Ca^2+^ in prokaryotes is largely unknown. Excess Ca^2+^ concentrations in the cytoplasm are toxic for both eukaryotes and prokaryotes. *Synechocystis* possesses a Ca^2+^/H^+^ antiporter in the plasma membrane, SynCax [64]. Deletion of synCax (Δ*syncax*) reduces Ca^2+^ efflux activity. In addition, the mutant has higher intracellular Na^+^ content and reduced growth in high salt medium. SynCax-mediated Ca^2+^ efflux across the cell membrane is related to adaptation to salinity stress.

## 5. Physiological Functions of CPAs in Plants

### 5.1. CPAs in the Plasma Membrane Are Required for Salt Extrusion and Nutrient Uptake

Plant CPAs are classified into eight Na^+^/H^+^ exchangers (NHX), 28 cation/H^+^ exchangers (CHX), six K^+^ efflux antiporters (KEA) and two sodium hydrogen antiporter (NHAD)-type carriers and one Ca^2+^/H^+^ exchanger (Table 3 and Figure 2C) [19]. In plant cells, CPAs localized in the cell membrane expel excess cations from the cell or replenish cations into the cell. *Arabidopsis thaliana* possesses a salt extrusion system of Na^+^/H^+^ antiporters, such as SOS1 (NHX7) [14]. The SOS (Salt Overly Sensitive) pathway consists of SOS1, SOS2, a CBL-interacting protein kinase (also called CIPK24), and SOS3, a calcineurin B-like protein (also called CBL4) [65]. When plant cells experience salt stress, the intracellular Ca^2+^ levels transiently increase. The binding of Ca^2+^ to SOS3 activates SOS2, resulting in phosphorylation of SOS1 and Na^+^ extrusion from the cytosol. Arabidopsis ∆*sos*1 accumulates Na^+^ in the cell, which inhibits the K^+^ uptake channel AKT1, leading to a decrease in the intracellular K^+^ concentration [66]. The Ca^2+^ for SOS activation may be replenished through a Ca^2+^ influx channel regulated by a glucuronosyltransferase (MOCA1) for glycosyl inositol phosphorylceramide (GIPC) sphingolipids in the plasma membrane [67]. NHX8 is also localized to the plasma membrane but is not involved in NaCl tolerance [68]. Growth of Arabidopsis ∆*nhx*8 is strongly reduced in the presence of 10 mM LiCl. This Li^+^ sensitivity is also seen in ∆*sos*1, and overexpression of NHX8 rescues the Li^+^ sensitivity of ∆*sos*1. Under high Li^+^ conditions, plants use SOS1 and NHX8 to extrude Li^+^ out of the cells.

Both CHX13 and CHX14 function as K^+^/H^+^ antiporters in the cell membrane, but their roles are different [69,70]. CHX13 has K^+^ uptake activity and therefore ∆*chx13* shows reduced growth and pale yellow leaves when grown on low K^+^ medium. On the other hand, CHX14 has K^+^ efflux activity and ∆*chx14* shows reduced root elongation and increased K^+^ content in root cells when exposed to excess K^+^.

### 5.2. Vacuolar CPAs Are Important for Cell Turgor Pressure and Storage of Intracellular K^+^

NHX1–NHX4 are localized in the vacuolar membrane and are involved in the regulation of cell turgor pressure, intracellular K^+^ accumulation and cytoplasmic Na^+^ homeostasis. NHX1 and NHX2 are specific for K^+^ over Na^+^ in their cation transport activity [71,72]. An Arabidopsis ∆*nhx*1 mutant shows reduced growth [71], while ∆*nhx*2 exhibits the same growth phenotype as the wild type. However, growth of the ∆*nhx1* ∆*nhx2* double mutant is more reduced than that of the ∆*nhx*1 single mutant. The ∆*nhx1* ∆*nhx2* double mutant also has smaller cells and immature development of embryos and stamen. NHX1-mediated K^+^ and Na^+^ transport activity is regulated by calmodulin (CaM)-like protein 15 (CAM15) [73]. CAM15 binds to the C-terminus of NHX1 inside the vacuole and significantly reduces Na^+^ transport activity of NHX1. CAM15 binds to NHX1 in a pH-dependent manner and fails to bind NHX1 under alkaline conditions. When plants are exposed to salt stress, the pH of the vacuole lumen increases [74]. Under non-stress conditions, NHX1 interacting with CAM15 may lead to swelling of the cell due to the influx of K^+^ into the vacuole. When the plant is exposed to salt stress, CAM15 separates from NHX1 and then NHX1 promotes Na^+^ transport into the vacuole.

NHX3 and NHX4 function differs from NHX1 and NHX2. NHX3 has a role in K^+^ accumulation in cells [75,76] and the expression of Arabidopsis *NHX3* in sugar beets increases salt stress tolerance [75]. *NHX3*-expressing sugar beets contain lower Na^+^ and higher K^+^ concentrations. In the same plants, K^+^/H^+^ antiporter activity in vacuolar membranes increases, but Na^+^/H^+^ antiporter activity does not. In Arabidopsis, *NHX3* is expressed in germinating seeds and reproductive tissues [76]. Arabidopsis ∆*nhx3* is sensitive to low K^+^ concentrations during germination, therefore NHX3 is likely involved in K^+^ uptake during seed germination. Tonoplast-localized NHX4 has Na^+^ transport activity, and translocates Na^+^ from the vacuole into the cytoplasm [77].

### 5.3. Regulation of Photosynthesis by Multiple CPAs

In chloroplasts, the light reaction leads to proton accumulation in the thylakoid lumen, resulting in the generation of a pH gradient component (∆pH) and membrane potential (∆Ψ). ATP production is driven by H^+^-ATPase using the proton motive force, which is the sum of ∆pH and ∆Ψ. Excessive reduction elicited by high light energy causes overproduction of reactive oxygen species, which hamper the photochemical system. To avoid this damage, non-photochemical quenching (NPQ) dissipates the collected light energy [78]. NPQ is triggered by acidification of the thylakoid lumen. Under low light conditions, NPQ is inhibited. KEA3 is a thylakoid lumen-localized K^+^/H^+^ antiporter. KEA3-mediated proton translocation from lumen to stroma inhibits NPQ [21,79,80,81,82,83]. NPQ is strongly suppressed in transgenic Arabidopsis with enhanced KEA3 transport activity [21,82]. Under constant light conditions the NPQ level of Arabidopsis ∆*kea3* is similar to the wild type, but the NPQ level of ∆*kea3* is increased under fluctuating light conditions [79,80]. KEA3 has a KTN domain in its C-terminal region and overexpression of a splicing variant without this domain represses NPQ. Interestingly, hyperosmotic stress does not lead to increased cytosolic Ca^2+^ concentrations in ∆*kea3*, suggesting that KEA3 is involved in Ca^2+^ release from the thylakoid lumen [84].

KEA1 and KEA2 function as regulators of photosynthesis in the envelope membrane of plastids [79,85]. While ∆*kea1* and ∆*kea2* do not show any changes in growth and photosynthesis, the double mutant ∆*kea1* ∆*kea2* shows impaired growth with pale leaves and reduced NPQ in high light. KEA1 and KEA2 contribute to maintaining K^+^ and H^+^ homeostasis in the chloroplasts, leading to correct induction of NPQ. NHD1, a sodium hydrogen antiporter (NHAD)-type carrier, also localizes to the chloroplast envelope [86]. NHD1 excludes Na^+^ from the chloroplast into the cytosol, protecting photosynthetic activity from salinity stress. In *Fraveria bidentis*, NHD1 establishes a Na^+^ gradient across the chloroplast envelope to activate Na^+^-dependent pyruvate uptake into the chloroplast via bile acid:sodium symporter family protein 2 (BASS2) [87].

The thylakoid lumen in the chloroplast is known to function as a Ca^2+^ store for stress responses [88]. In addition, the oxygen evolving complex (OEC) in the thylakoid membrane requires Ca^2+^. Ca^2+^ transport across the thylakoid membrane is crucial for chloroplasts, however the transport mechanism is unknown. The chloroplast-localized Ca^2+^/H^+^ antiporter CCHA1 is a potential candidate for this Ca^2+^ transport system into the thylakoid lumen [89,90]. The *ccha*1 mutation in Arabidopsis decreases photosynthetic activity and oxygen generation rate and alters the cytoplasmic Ca^2+^ concentration and the pH in guard cells. CCHA1 likely regulates photosynthetic activity and stomatal aperture by exchanging Ca^2+^ and H^+^ across the thylakoid membrane.

### 5.4. Role of Endosome-Localized CPAs

Various types of CPAs are localized in endosomal membranes, functioning in regulating the internal ion concentration and pH. NHX5 and NHX6 participate in control of the endosomal ion environment [18,91,92]. The acidic pH in the trans-Golgi network (TGN), prevacuolar compartment (PVC) and vacuolar sorting receptors (VSRs) in ∆*nhx*5∆*nhx*6 suggests that NHX5 and NHX6 transport K^+^ into endosomes and release H^+^ into the cytoplasm [91]. *NHX6* is expressed in primary root tips and lateral root primordia. Delayed growth of lateral root primordia occurs in ∆*nhx*5∆*nhx*6 [92]. Moreover, transport activity of the auxin efflux PIN FORMED (PIN) family, which is required for correct auxin distribution, is inhibited in the plasma membrane of ∆*nhx*5∆*nhx*6. NHX5 and NHX6 therefore promote salt extrusion and auxin distribution.

CHX17 is localized in the PVC [93], and its C-terminal region is important for this localization [94]. Expression of *CHX17* is enhanced by salt stress, K^+^ starvation, low pH and ABA treatment. Salt stress decreases K^+^ content in ∆*chx*17. CHX17/18/19 localize to the PVC, and CHX20 localizes to both PVC and vacuolar membrane [17]. CHX17 and CHX20 are involved in protein sorting. CHX23 localizes to the endoplasmic reticulum (ER) in pollen tubes [95]. Simultaneous deletion of *CHX23* and plasma membrane-localized *CHX21* reduces pollen tube guidance to ovules.

KEA4/5/6 is expressed in the Golgi and in endosomes in all plant tissues [96,97]. A triple mutant of ∆*kea4/5/6* has reduced growth, and is highly sensitive to K^+^ starvation and salinity stress [96]. The ∆*kea4/5/6* triple mutant has acidic pH in organelles, similar to ∆*nhx5* and ∆*nhx6*. Expression of *NHX5* or *NHX6* rescues the salt stress sensitivity of ∆*kea4/5/6*, whereas expression of the plasma membrane transporters *NHX1* and *NHX2* cannot rescue the mutant. The function of KEA4/5/6 is similar to that of NHX5/6 [97].

## 6. Conclusions

Numerous reports show the structure and function of CPAs and physiological importance of the exchange of cations and H^+^ across membranes in living cells. The transport of cations and H^+^ by CPAs directly controls Na^+^/K^+^ and proton concentrations in the cytosol, and indirectly supports photosynthesis, protein processing and resistance against toxins. Further studies are needed to fully understand the transport mechanism of CPAs and their physiological role in cellular signaling pathways that contribute to improved plant tolerance to abiotic and biotic stress.

## Figures and Tables

**Figure 1 ijms-21-04566-f001:**
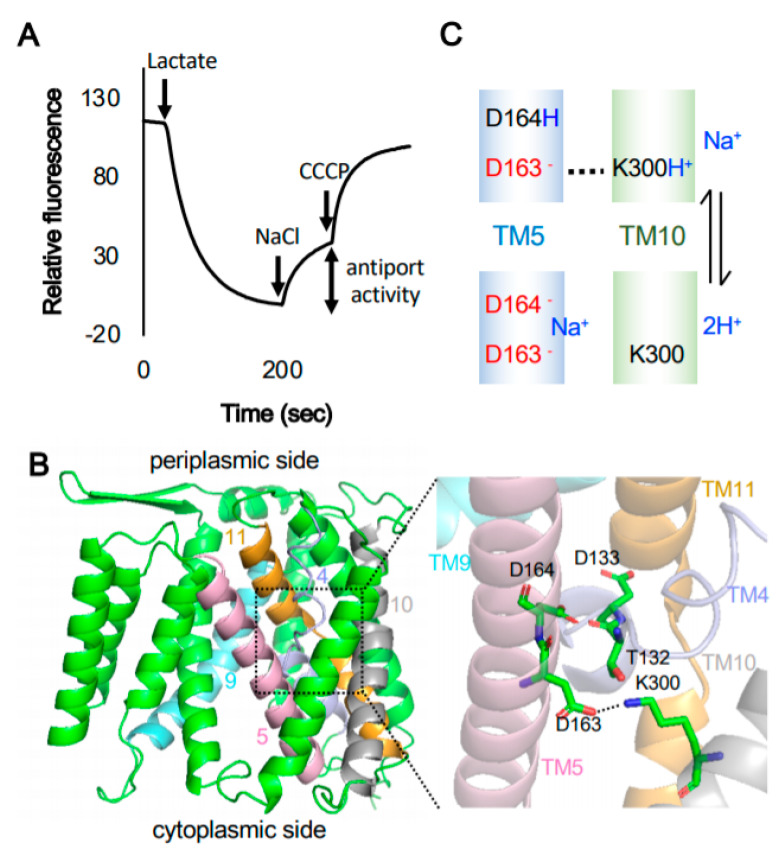
Structure and molecular function of NhaA. (**A**) Typical profile of the proton movement across an inverted membrane vesicle. Na^+^/H^+^ antiporter activity is monitored by dequenching of a pH sensitive fluorescent probe like acridine orange. (**B**) Three-dimensional structure model of NhaA, the Na^+^/H^+^ antiporter of *E. coli* (PDB 4AU5 [26]), was made by using PyMOL, version 2.3.5, Schrodinger LLC. Overview (left) and ion biding sites (right). The numbers in the left figure correspond to the numbers of the transmembrane domains. (**C**) Proposed mechanism of Na^+^ and H^+^ translocation by NhaA [26], D163 and K300 form a salt bridge (dotted line). When Na^+^ binds to D163 and D164, the salt bridges are disrupted, and two protons are released from D163 and K300.

**Figure 2 ijms-21-04566-f002:**
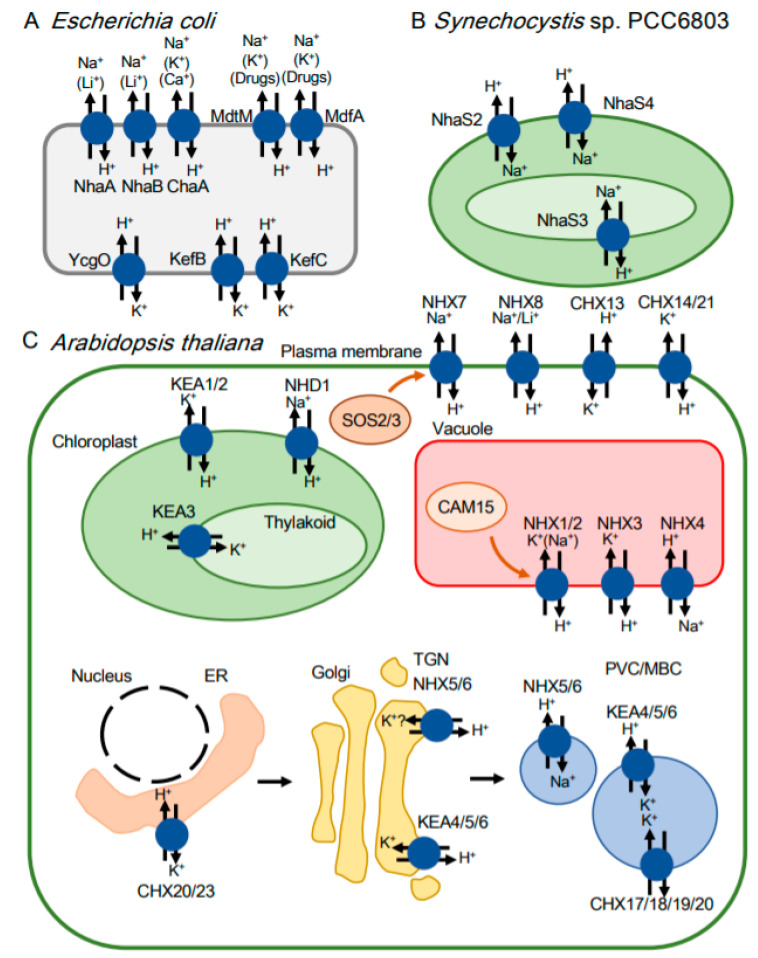
Localization of CPAs (cation/proton antiporters) in *Escherichia coli* (**A**), cyanobacteria (**B**) and *Arabidopsis thaliana* (**C**).

**Table 1 ijms-21-04566-t001:** CPAs in *Escherichia coli.*

Transporter	Activity	Physiological Function	Family	Reference
NhaA	Na^+^(Li^+^)/H^+^ antiporter	Na^+^ extrusion, alkaline pH homeostasis	NhaA	(Taglicht et al., 1991)
NhaB	Na^+^(Li^+^)/H^+^ antiporter	Na^+^ extrusion, alkaline pH homeostasis	NhaB	(Pinner et al., 1993, Shimamoto et al., 1994)
ChaA	Na^+^(Ca^+^, K^+^)/H^+^ antiporter	Na^+^ extrusion, alkaline pH homeostasis, KCl salinity tolerance	CaCA	(Ivey et al., 1993, Radchenko et al., 2006)
KefB	glutathione-regulated K^+^ efflux (K^+^/H^+^ antiporter)	electrophilic stress tolerance	CPA2	(Ferguson et al., 1997, Elmore et al., 1997)
KefC	glutathione-regulated K^+^ efflux (K^+^/H^+^ antiporter)	electrophilic stress tolerance	CPA2	(Ferguson et al., 1997, Elmore et al., 1997)
MdfA	drugs (Na^+^,K^+^)/H^+^ antiporter	antimicrobial resistance, alkaline pH homeostasis	DHA1	(Lewinson et al., 2004, Lewinson et al., 2003)
MdtM	drugs (Na^+^,K^+^, Rb^+^, Li^+^)/H^+^ antiporter	antimicrobial resistance, alkaline pH homeostasis	DHA1	(Holdsworth et al., 2013)
YcgO	K^+^/H^+^ antiporter	growth at low osmolarity	CPA1	(Sharma et al., 2016, Verkhovskaya et al., 2001)

**Table 2 ijms-21-04566-t002:** CPAs in cyanobacteria.

Prokaryote Isoform	Membrane Location	Activity	Physiological Function	Family	Reference
NhaS1 (*Syncechocystis* sp. pcc6803)	-	Na^+^ (Li^+^)/H^+^ antiporter	-	CPA1	(Inaba et al., 2001, Hamada et al., 2001, Waditee et al., 2006)
NhaS2 (*Syncehocystis* sp. pcc6803)	-	-	Na^+^ influx	CPA1	(Mikkat et al., 2000, Wang et al., 2002)
NhaS3 (*Syncehocystis* sp. pcc6803)	Thylakoid membrane	Na^+^/H^+^ antiporter	Na^+^ homeostasis, adaptation to osmotic stress	CPA2	(Wang et al., 2002, Tsunekawa et al., 2009)
NhaS4 (*Syncehocystis* sp. pcc6803)	-	-	Na^+^ influx	CPA2	(Wang et al., 2002)
SynCAX (*Syncehocystis* sp. pcc6803)	-	Ca^2+^/H^+^ antiporter	alkaline homeostasis, adaptation to salt stress	-	(Waditee et al., 2004)
NhaS2 (*Syncechococcus* sp. strain PCC 7942)	-	-	Na^+^ influx	CPA1	(Billini et al., 2008)
NhaS3 (*Syncechococcus* sp. strain PCC 7942)	-	Na^+^/H^+^ antiporter	Na^+^ homeostasis	CPA2	(Billini et al., 2008)

**Table 3 ijms-21-04566-t003:** CPAs in Arabidopsis thaliana.

Plant Isoform	Membrane Location	Activity	Physiological Function	Family	Reference
NHX1, NHX2	Vacuole	K^+^(Na^+^)/H^+^ antiporter	maintaining cell turgor pressure, adaptation to excess K^+^ conditions, flowering	CPA1	(Shi and Zhu, 2002, Bassil et al., 2011, Barragan et al., 2012)
NHX3	Vacuole	K^+^(Na^+^)/H^+^ antiporter	adaptation to low K^+^ conditions	CPA1	(Liu et al., 2008, Liu et al., 2010)
NHX4	Vacuole	Na^+^/H^+^ antiporter	Na^+^ homeostasis	CPA1	(Li et al., 2009)
NHX5, NHX6	Golgi, Trans-golgi network, prevacuolar compartment	-	protein sorting to vacuole, adaptation to salt stress, root extention, redistribution of auxin	CPA1	(Bassil et al., 2011, Reguela et al., 2015, Dragwidge et al., 2018)
NHX7 (SOS1)	Plasma membrane	Na^+^/H^+^ antiporter	Na^+^ extrusion, maintaining intracellular K^+^ content	CPA1	(Wu et al., 1996, Shi et al., 2002, Qiu et al., 2002, Qi and Spalding, 2004, Ariga et al., 2013)
NHX8	Plasma membrane	(Li^+^)/H^+^ antiporter	Li^+^ extrusion	CPA1	(An et al., 2007)
CHX13	Plasma membrane	K^+^/H^+^ antiporter	K^+^ uptake	CPA2	(Zhao et al., 2008)
CHX14	Plasma membrane	K^+^/H^+^ antiporter	K^+^ extrusion	CPA2	(Zhao et al., 2015)
CHX17	Prevacuolar compartment	K^+^/H^+^ antiporter	protein sorting to vacuole, K^+^ homeostasis, seed development	CPA2	(Cellier et al., 2004, Chanroj et al., 2011, Chanroj et al., 2013)
CHX18, CHX19	Prevacuolar compartment	K^+^/H^+^ antiporter	-	CPA2	(Chanroj et al., 2011)
CHX20	Prevacuolar compartment, ER	K^+^/H^+^ antiporter	protein sorting to vacuole	CPA2	(Chanroj et al., 2011)
CHX21	Plasma membrane	-	adaptation to high Na^+^ (K^+^) conditions, pollen tube growth	CPA2	(Hall et al., 2006, Evans et al., 2011)
CHX23	ER	K^+^/H^+^ antiporter	pollen tube growth	CPA2	(Lu et al., 2011, Evans et al., 2011)
KEA1, KEA2	Plastid envelope	K^+^/H^+^ antiporter	chloroplast development, adaptation to hyper-osmotic stress	CPA2	(Kunz et al., 2014, Aranda Sicilia et al., 2016, Stephan et al., 2016, Tsujii et al., 2019)
KEA3	Plastid thylakoid membrane	K^+^/H^+^ antiporter	fine-tuning of photosynthesis, chloroplast development, adaptation to hyper-osmotic stress	CPA2	(Kunz et al., 2014, Armbruster et al., 2014, 2016, Stephan et al., 2016, Wang et al., 2017, 2019, Tsujii et al., 2019)
KEA4, KEA5, KEA6	Golgi, Trans-golgi network, prevacuolar compartment	K^+^/H^+^ antiporter	maintaining ion homeostasis in low K^+^ conditions, high K^+^(Na^+^) stress, pH regulation in vacuole	CPA2	(Zhu et al., 2018, Wang et al., 2018, Tsujii et al., 2019)
NHD1	Plastid envelope	Na^+^/H^+^ antiporter	Na^+^ extrusion from chloroplast	NhaD	(Maria et al., 2014)
CCHA1 (PAM71)	Plastid thylakoid membrane	-	Photosynthetic regulation, regulation of cytosolic pH in guard cells	-	(Wang et al., 2016, Schneider et al., 2016)

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
