# Peer review of "Diverse Physiological Functions of Cation Proton Antiporters across Bacteria and Plant Cells"

_ijms, 2020, doi:10.3390/ijms21124566_

Round 1
Reviewer 1 Report
This review of the functioning of cation-proton antiporters in bacteria and plants is well constructed, comprehensive and very well written. It provides a good discussion of current knowledge and understanding of the existence and roles of cation-proton antiporters in bacteria and plants. The manuscript clearly identifies topics that are well researched and ones that require further investigation.
Unfortunately, the document available to me did not included the figures. Once I have been able to review these figures and subject to these being of a suitable quality and standard for publication, I would wholeheartedly recommend that this manuscript be accepted for publication after the following very minor changes have been made.
L 24 replace "osmolarity" with "osmotic"
L 25 insert a space between "plasma" and "membrane"
L 75 replace "henceforward" with "henceforth"
L 80 replace "thereby" with "therefore"
L 101 remove ". " after "D163" and insert "which suggests" to replace "gives"
L 155 replace "extrud" with "exported"
L 156 as per L 155
L 176 insert "the" before "case"
L 178 insert "the" before "case"
L 216 replace "medium" with "media"
L 239 replace "undergo" with "experience"
L 243 remove "in" from "to in a"
L 251 add "s" to "antiporter"
L 299 remove "," after "domain"
L 305 replace "maintain" with "maintaining"
Figures:
The figures look good, but nuclei in Figure 2 c should be changed to nucleus.
Author Response
Response to Reviewer 1's Comments
I would wholeheartedly recommend that this manuscript be accepted for publication after the following very minor changes have been made.
Thank you very much for your positive feedback. We have made all of the suggested changes in the text and in Figure 2C.
Reviewer 2 Report
The manuscript by Tsujii et al is a very interesting review on the functions of cation proton antiporters from bacteria and plant. The manuscript is well presented and organized even if some concerns arose which are listed below:
1) As a general comment, the two mentioned figures are missing.
2) In the introduction:
- line 47: please give some information on concentrations to validate the written assumption.
- line 53: The introduction on CPA should be connected to the previous information on Na+ and K+ and Na+K+ pumps.
3) In the paragraph 1:
- lines 83-84: This sentence is unclear (has the antiporter only 2 TM segments?).
- line 101: typo.
- line 119: here the sequence should be provided with a multiple alignement of the cited transporters.
- line 120: The issue of the electrogenicity should be more clearly discussed as deriving from Na exchange for 2 protons.
4) In the paragraph 2.1, line 139: a figure would help.
5) In the paragraph 2.2, line 157: the issue of electrogenicity should be addressed.
6) In the paragraph 2.3, line 175: Some words on the specificity towards drugs (cations?) should be added for clarity.
7) In the paragraph 4.1, line 232: A speculation on similarity/homology between plant and cyanobacetria with alignments, conserved residues etc should be provided.
8) Conclusions should be improved with a broader perspective on potential outcome for plant resistance.
Author Response
Response to Reviewer 2's Comments
1) As a general comment, the two mentioned figures are missing.
Thank you very much for reviewing our manuscript. We included two figures with our initial submission. I have resubmitted the figures and they should be available to you.
2) In the introduction:
- line 47: please give some information on concentrations to validate the written assumption.
We have added the following sentence to explain the concentrations of potassium and sodium. “The concentration of K+ in the cells is estimated to be around 0.1-0.2 M and it is the most prevalent cation in living cells [1,2].” “In contrast, the Na+ concentration in cells is usually only 5-14 mM.” (Line 46-51)
- line 53: The introduction on CPA should be connected to the previous information on Na+ and K+ and Na+/K+ pumps.
We have added “… since a Na+/K+-ATPase is missing in bacteria and plants”. (Line 53)
- lines 83-84: This sentence is unclear (has the antiporter only 2 TM segments?).
We have amended this sentence. The monomeric crystal structure of NhaA reveals a unique structural conformation consisting of TM4 and TM11 [24], designated as the NhaA fold [26]. These two transmembrane regions contain the ion-binding site at their intersection [26,27]. (Line 85-86)
- line 101: typo.
Thank you very much. We have removed the unnecessary period. (Line 102)
- line 119: here the sequence should be provided with a multiple alignement of the cited transporters.
We agree that providing a multiple sequence alignment can be useful to compare the sequences of the mentioned transporters. However, the cited reference [26] already summarizes sequence identity and similarity of these transporters (based on a multiple sequence alignment), therefore we think that it is not necessary to also include in our manuscript.
- line 120: The issue of the electrogenicity should be more clearly discussed as deriving from Na exchange for 2 protons.
As we explained in the paragraph above (Line 97- Line 113), the exact transport mechanism of NhaA is under discussion and the possible electrogenicity of NhaA and CPA remains to be investigated.
4) In the paragraph 2.1, line 139: a figure would help.
Figure 2A depicts all Na+ extrusion systems in E. coli. (Line140)
5) In the paragraph 2.2, line 157: the issue of electrogenicity should be addressed.
The original papers did not explain the effect of electrogenicity on the physiological role involving proton motive force (pH gradient and membrane potential), and the issue of the electrogenicity is not fully understood. Thus, we would prefer to not address the topic of electrogenicity in any more detail.
6) In the paragraph 2.3, line 175: Some words on the specificity towards drugs (cations?) should be added for clarity.
MdtM and MdfA extrud a broad range of toxic compounds like antibiotics, mutagen and xenobiotics. We have added “such as antibiotics, mutagen and xenobiotics” in the text. We have added the We have specified more clearly which transporter, MtdA or MdfA, is involved in the extrusion of which drug. (Line 176-179)
7) In the paragraph 4.1, line 232: A speculation on similarity/homology between plant and cyanobacetria with alignments, conserved residues etc should be provided.
We agree that providing similarity/homology is useful for the reader. However, a phylogenetic tree and conserved motif analysis of CPAs for plants and bacteria has already been provided in reference [18]. We therefore think that it is not necessary to also include this in our manuscript. (Line233)
8) Conclusions should be improved with a broader perspective on potential outcome for plant resistance.
According to the reviewer’s suggestion, we have added the sentence “that contribute to improved plant tolerance to abiotic and biotic stress” (Line355-356).